computer-aided design/artificial intelligence/computer vision

deep learning, urban design, generative models, urban beauty, explainable models

**Author for correspondence:**
Sagar Joglekar
e-mail: sagar.joglekar@nokia-bell-labs.com

# FaceLift: a transparent deep learning framework to beautify urban scenes

Sagar Joglekar[1,2], Daniele Quercia[1,2], Miriam Redi[2], Luca Maria Aiello[2], Tobias Kauer[2] and Nishanth Sastry[1]

[1]King's College, London, UK
[2]Nokia Bell labs, Cambridge, UK

SJ, 0000-0002-8388-9137; LMA, 0000-0002-0654-2527

In the area of computer vision, deep learning techniques have recently been used to predict whether urban scenes are likely to be considered beautiful: it turns out that these techniques are able to make accurate predictions. Yet they fall short when it comes to generating actionable insights for urban design. To support urban interventions, one needs to go beyond *predicting* beauty, and tackle the challenge of *recreating* beauty. Unfortunately, deep learning techniques have not been designed with that challenge in mind. Given their 'black-box nature', these models cannot be directly used to explain why a particular urban scene is deemed to be beautiful. To partly fix that, we propose a deep learning framework (which we name FaceLift[1]) that is able to both *beautify* existing urban scenes (Google Street Views) and *explain* which urban elements make those transformed scenes beautiful. To quantitatively evaluate our framework, we cannot resort to any existing metric (as the research problem at hand has never been tackled before) and need to formulate new ones. These new metrics should ideally capture the presence (or absence) of elements that make urban spaces great. Upon a review of the urban planning literature, we identify *five* main metrics: walkability, green spaces, openness, landmarks and visual complexity. We find that, across all the five metrics, the beautified scenes meet the expectations set by the literature on what great spaces tend to be made of. This result is further confirmed by a 20-participant expert survey in which FaceLift has been found to be effective in promoting citizen participation. All this suggests that, in the future, as our framework's components are further researched and become better and more sophisticated, it is not hard to imagine technologies that will be able to accurately and efficiently support architects and planners in the design of the spaces we intuitively love.

[1]The project's website is goodcitylife.org/facelift/.

# 1. Introduction

Whether a street is considered beautiful is subjective, yet research has shown that there are specific urban elements that are universally considered beautiful: from greenery, to small streets, to memorable spaces [1–3]. These elements are those that contribute to the creation of what the urban sociologist Jane Jacobs called 'urban vitality' [4].

Given that, it comes as no surprise that computer vision techniques can automatically analyse pictures of urban scenes and accurately determine the extent to which these scenes are considered, *on average*, beautiful. Deep learning has greatly contributed to increase these techniques' accuracy [5].

However, urban planners and architects are interested in urban interventions and, as such, they would welcome machine learning technologies that help them recreate beauty in urban design [6] rather than simply predicting beauty scores. As we shall see in §2, deep learning, by itself, is not fit for purpose. It is not meant to recreate beautiful scenes, not least because it cannot provide any explanation on why a scene is deemed beautiful, or which urban elements are predictors of beauty.

To partly fix that, we propose a deep learning framework (which we name FaceLift) that is able to both *generate* a beautiful scene (or, better, *beautify* an existing one) and *explain* which parts make that scene beautiful. Our work contributes to the field of urban informatics, an interdisciplinary area of research that studies practices and experiences across urban contexts and creates new digital tools to improve those experiences [7,8]. Specifically, we make two main contributions:

— We propose a deep learning framework that is able to learn whether a particular set of Google Street Views (urban scenes) are beautiful or not, and based on that training, the framework is then able to both *beautify* existing views and *explain* which urban elements make them beautiful (§3).
— We quantitatively evaluate whether the framework is able to actually produce beautified scenes (§4). We do so by proposing a family of five urban design metrics that we have formulated based on a thorough review of the literature in urban planning. For all these five metrics, the framework passes with flying colours: with minimal interventions, beautified scenes are twice as walkable as the original ones, for example. Also, after building an interactive tool with 'FaceLifted' scenes in Boston and presenting it to 20 experts in architecture, we found that the majority of them agreed on three main areas of our work's impact: decision making, participatory urbanism and the promotion of restorative spaces.

# 2. Related work

Previous work has focused on: collecting ground truth data about how people perceive urban spaces; predicting urban qualities from visual data; and generating synthetic images that enhance a given quality (e.g. beauty).

## 2.1. Perception of physical spaces

From Jane Jacobs's seminal work on urban vitality [4] to Christopher Alexander's cataloguing of typical 'patterns' of good urban design [1], there has been a continuous effort to understand what makes our cities liveable and enjoyable. In the fields of psychology, environmental design and behavioural sciences, research has studied the relationship between urban aesthetics [9] and a variety of objective measures (e.g. scene complexity [10], the presence of nature [11]) and subjective ones (e.g. people's affective responses [12]).

## 2.2. Ground truth of urban perceptions

So far, the most detailed studies of perceptions of urban environments and their visual appearance have relied on personal interviews and observation: some researchers relied on annotations of video recordings by experts [13], while others have used participant ratings of simulated (rather than existing) street scenes [14]. The Web has recently been used to survey a large number of individuals. Place Pulse is a website that asks a series of binary perception questions (such as 'Which place looks safer [between the two]?') across a large number of geo-tagged images [3]. In a similar way, Quercia *et al.* collected pairwise judgements about the extent to which urban scenes are considered quiet, beautiful and happy [2] to then recommend pleasant paths in the city [15]. They were then able to analyse the scenes together with their ratings using image-processing tools, and found that the

amount of greenery in any given scene was associated with all three attributes and that cars and fortress-like buildings were associated with sadness. Taken all together, their results pointed in the same direction: urban elements that hinder social interactions were undesirable, while elements that increase interactions were the ones that should be integrated by urban planners to retrofit cities for happiness. Urban perceptions translate in concrete outcomes. Based on 3.3k self-reported survey responses, Ball *et al.* [16] found that urban scenes that are aesthetically beautiful not only are visually pleasurable but also promote walkability. Similar findings were obtained by Giles-Corti *et al.* [17].

## 2.3. Deep learning and the city

Computer vision techniques have increasingly become more sophisticated. Deep learning techniques, in particular, have been recently used to accurately predict urban beauty [5,18], urban change [19] and even crime [20,21]. Recent works have also shown the utility of deep learning techniques in predicting house prices from urban frontages [22], and from a combination of satellite data and street view images [23].

## 2.4. Generative models

Since the introduction of generative adversarial networks (GANs) [24], deep learning has been used not only to analyse existing images but also to generate new ones altogether. This family of deep networks has evolved into various forms, from super-resolution image generators [25] to fine-grained in-painting technologies [26]. Recent approaches have been used to generate images conditioned on specific visual attributes [27], and these images range from faces [28] to people [29]. In a similar vein, Nguyen *et al.* [30] used generative networks to create a natural-looking image that maximizes a specific neuron (the beauty neuron). In theory, the resulting image is the one that 'best activates' the neuron under consideration. In practice, it is still a synthetic template that needs further processing to look realistic. Finally, with the recent advancement in augmented reality, the application of GANs to generate urban objects in simulated urban scenes have also been attempted [31].

To sum up, a lot of work has gone into collecting ground truth data about how people tend to perceive urban spaces, and into building accurate predictions models of urban qualities. Yet little work has gone into models that generate realistic urban scenes that maximize a specific property and that offer human-interpretable explanations of what they generate.

# 3. FaceLift framework

The main goal of FaceLift is to beautify an existing urban scene and explain its beautification. To meet that goal, it performs five steps:

**Step 1: Curating urban scenes.** Deep learning systems need considerable amounts of training data. Our initial set of data is limited, and, to augment it, we develop a new way of curating and augmenting the number of annotated images.

**Step 2: Training a beauty classifier.** We design and train a deep learning model that is able to distinguish beautiful urban scenes from non-beautiful ones.

**Step 3: Generating a synthetic beautified scene.** Based on our classifier's learned representation of beauty, we train a generative model that is able to beautify an urban scene in input.

**Step 4: Retrieving a realistic beautified scene.** The generated image has a 'synthetic look'—it does not look realistic. To fix that, we retrieve the image in the set of curated urban scenes most similar to the generated one. We use the Euclidean distance to compute similarity.

**Step 5: Identifying the urban elements characterizing the beautified scene.** In the final step, the framework explains the changes introduced in the transformation process by comparing the beautified scene to the original one in terms of addition and removal of specific urban elements. An end-to-end illustration of the Facelift framework can be seen in figure 1.

## 3.1. Step 1: Curating urban scenes

To begin with, we need highly curated training data with labels reflecting urban beauty. We start with the Place Pulse dataset that contains a set of 110 000 Google Street View images from 56 major cities across 28 countries around the world [5]. The pictures were labelled by volunteers through an *ad hoc* crowdsourcing website.[2]

---

[2]See http://pulse.media.mit.edu.

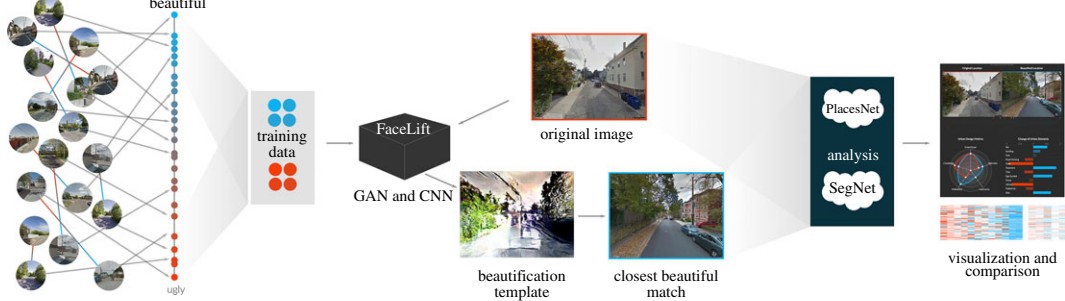

**Figure 1.** An illustration of the FaceLift framework. The training data of urban scenes with beautiful and ugly labels is fed into the FaceLift algorithms (the generative adversarial network and convolutional neural network models). These transform the original image into its beautified version. The two images—original and beautified—are compared in terms of urban elements that have been added or removed.

Volunteers were shown random pairs of images and asked to select which scene looked more beautiful, safe, lively, boring, wealthy and depressing. At the time of writing, 1.2 million pairwise comparisons were generated by 82 000 online volunteers from 162 countries, with a good mix of people residing in both developed and developing countries. To our knowledge, no independent systematic analysis of the biases of Place Pulse has been conducted yet. However, it is reasonable to expect that representation biases are minimized by the substantial size of the dataset, the wide variety of places represented, and the diversity of gender, racial and cultural backgrounds of the raters. We focus only on those scenes that are labelled in terms of beauty (the focus of this study) and that have at least three judgements. This leave us with roughly 20 000 scenes. To transform judgements into beauty scores, we use the TrueSkill algorithm [32], which gives us a way of partitioning the scenes into two sets (figure 2): one containing beautiful scenes, and the other containing ugly scenes. The resulting set of scenes is too small for training any deep learning model without avoiding over-fitting though. As such, we need to augment such a set.

We do so in two ways. First, we feed each scene's location into the Google Street View API (application programming interface) to obtain the snapshots of the same location at different camera angles (i.e. at $\theta \in -30°, -15°, 15°, 30°$). Yet the resulting dataset is still too small for robust training. So we again feed each scene's location into the Google Street View API, but this time we do so to obtain scenes at increasing distance $d \in \{10, 20, 40, 60\}$ metres. A real example of one such augmentation instance can be seen in figure 3). This expands our set of scenes, but does so at the price of introducing scenes whose beauty scores have little to do with the original one's. To fix that, we take only the scenes that are *similar* to the original one (we call this way of augmenting 'conservative translation'). Two scenes are considered similar if the similarity of their two feature vectors (derived from the FC7 layer of PlacesNet [33]) is above a certain threshold. In a conservative fashion, we choose that threshold to be the median similarity between rotated and original scenes.

To make sure this additional augmentation has not introduced any unwanted noise, we consider two sets of scenes: one containing those that have been taken during this last step, i.e. the one with high similarity to the original scenes (*taken-set*), and the other containing those that have been filtered away (*filtered-set*). Each scene is represented with the five most confident scene labels extracted by PlacesNet [33]. We then aggregate labels at set level by computing each label's frequency fr on the *taken-set* and that on the *filtered-set*. Finally, we characterize each label's propensity to be correctly augmented as: prone (label) = fr(label, *taken-set*) − fr(label, *filtered-set*). This reflects the extent to which a scene with a given label is prone to be augmented or not. From figure 4, we find that, as one would expect, scenes that contain highways, fields and bridges can be augmented at increasing distances while still showing resemblances to the original scene; by contrast, scenes that contain gardens, residential neighbourhoods, plazas and skyscrapers cannot be easily augmented, as they are often found in high-density parts of the city in which diversity within short distances is likely to be experienced.

## 3.2. Step 2: Training a beauty classifier

Having this highly curated set of labelled urban scenes, we are now ready to train a classifier. We choose the CaffeNet architecture as our classifier *C*. This is a modified version of AlexNet [34,35]. Its Softmax layer classifies the input image into one of two classes of beautiful(1) and ugly(0).

Having *C* at hand, we now turn to training it. The training is done on a 70% split of each of the training sets, and the testing on the remaining 30%. The training sets are constructed as increasingly augmented sets of data. We start from our 20 000 images and progressively augment them with the

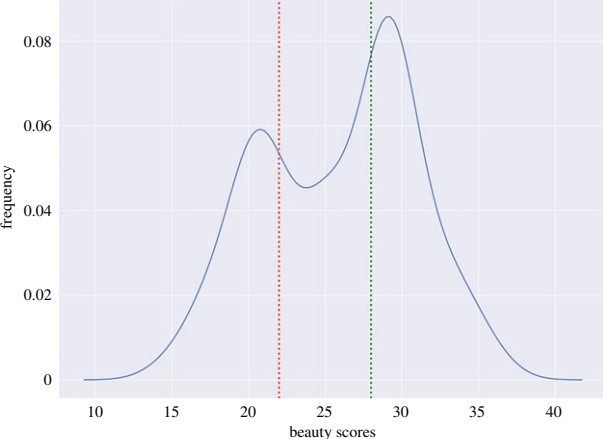

**Figure 2.** Frequency distribution of beauty scores. The red and green lines represent the thresholds below and above which images are considered ugly and beautiful. Conservatively, images in between are discarded.

(*a*)                                         (*b*)

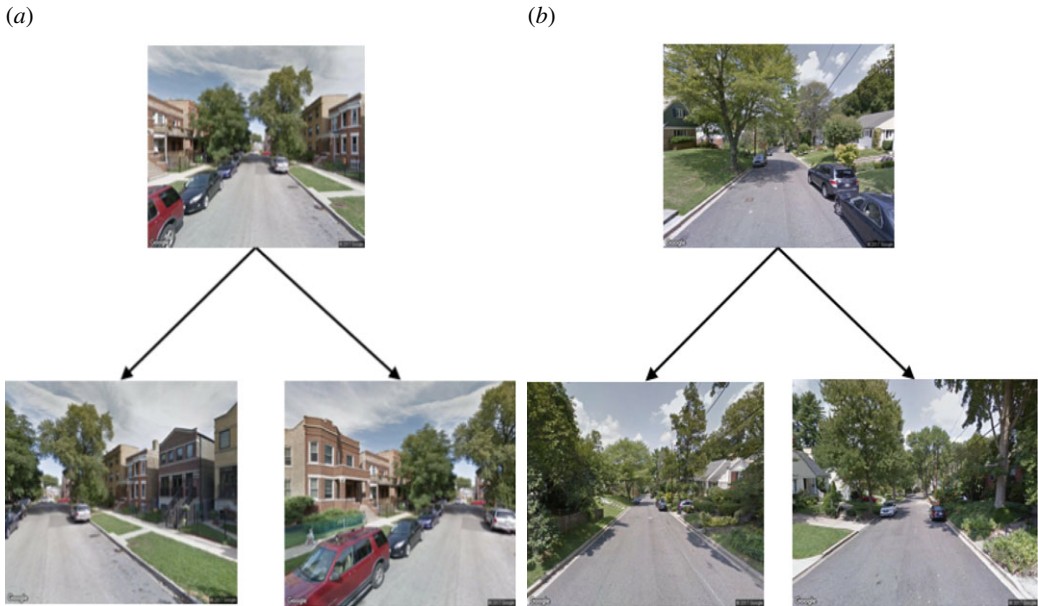

**Figure 3.** Two types of augmentation: (*a*) *rotation* of the Street Views camera and (*b*) *translation* of scenes at increasing distances.

snapshots obtained with the five-angle camera rotations, and then with the exploration of scenes at increasing distance $d \in \{10, 20, 40, 60\}$ metres. The idea behind this data augmentation is that the model's accuracy should increase with increasing levels of augmentation. Indeed it does (table 1): it goes from 63% on the set of original scenes to a value as high as 73.5% on the set of fully augmented scenes, which is a notable increase in accuracy for this type of classification tasks (the state-of-the-art classifier was reported to have an accuracy of 70% [5]).

## 3.3. Step 3: Generating a synthetic beautified scene

Having this trained classifier at hand, we can then build a generator of synthetic beautified scenes. To build such a generator, we retrain the generative adversarial network (GAN) described by Dosovitskiy & Brox [36] on our curated urban scene dataset (summary of terms can be found in table 2). This network is trained by maximizing the confusion for the discriminator between the generated images $G(f)$ and the original ones $I_f$ [24]. Some examples of the output of this generator can be seen in table 3. This table shows the comparison between the original $I_f$ and the GAN's generated image $G(f)$ side by side. The resulting generator is concatenated with our beauty classifier (figure 5). As a result, given the two classes of ugly $y_i$ and beautiful $y_j$, the end-to-end model transforms any original scene $I_i$ of class $y_i$ (e.g. ugly scene) into template scene $\hat{I}_j$ that maximizes class $y_j$ (e.g. beautified template scene).

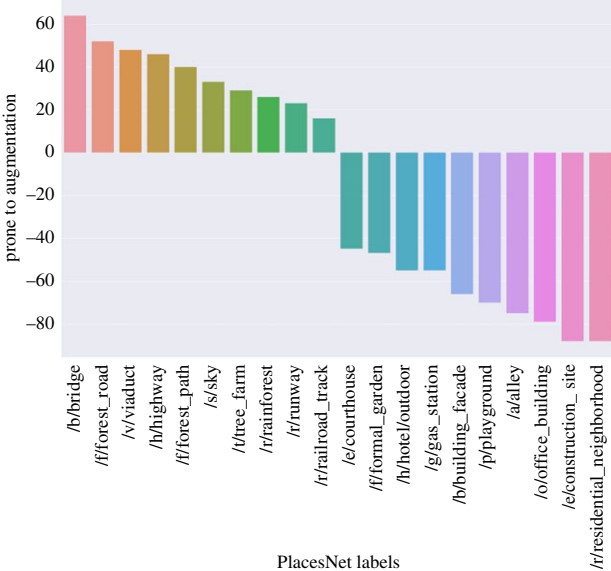

**Figure 4.** The types of scene that have greater propensity to be correctly augmented with similar scenes at increasing distances.

**Table 1.** Percentage accuracy for our beauty classifier trained on sets of urban scenes that have been differently augmented.

| augmentation | accuracy (percentage) |
| --- | --- |
| none | 63 |
| rotation | 68 |
| rotation + translation | 64 |
| rotation + conservative translation | 73.5 |

**Table 2.** Notation.

| symbol | meaning |
| --- | --- |
| $I_i$ | original urban scene |
| $Y$ | set of annotation classes for urban scenes (e.g. beautiful, ugly) |
| $y_i$ | annotation class in $Y$ (e.g. beautiful) |
| $\hat{I}_j$ | template scene (synthetic image) |
| $I'$ | target image |
| $C$ | beauty classifier |

More specifically, given an input image $I_i$ known to be of class $y_i$ (e.g. ugly), our technique outputs $\hat{I}_j$, which is a more beautiful version of it (e.g. $I_i$ is morphed towards the average representation of a beautiful scene) while preserving the way $I_i$ looks. The technique does so using the 'Deep Generator Network for Activation Maximization' (*DGN-AM*) [30]. Given an input image $I_i$, *DGN-AM* iteratively recalculates the colour of $I_i$'s pixels in a way the output image $\hat{I}_j$ both maximizes the activation of neuron $y_j$ (e.g. the 'beauty neuron') and looks 'photo realistic'. This is equivalent to finding the feature vector $f$ that maximizes the following expression:

$$\hat{I}_j = G(f) : \arg\max_f \left( C_j(G(f)) - \lambda \|f\| \right), \tag{3.1}$$

where $G(f)$ is the image synthetically generated from the candidate feature vector $f$; $C_j(G(f))$ is the activation value of neuron $y_j$ in the scene classifier $C$ (the value to be maximized); $\lambda$ is an $L_2$ regularization term.

Here, the initialization of $f$ is key. If $f$ were to be initialized with random noise, the resulting $G(f)$ would be the average representation of category $y_j$ (of e.g. beauty). Instead, $f$ is initialized with the

**Table 3.** Examples of our generator's outputs. The original scenes and the generated ones are shown side by side.

| original | generated |
| --- | --- |

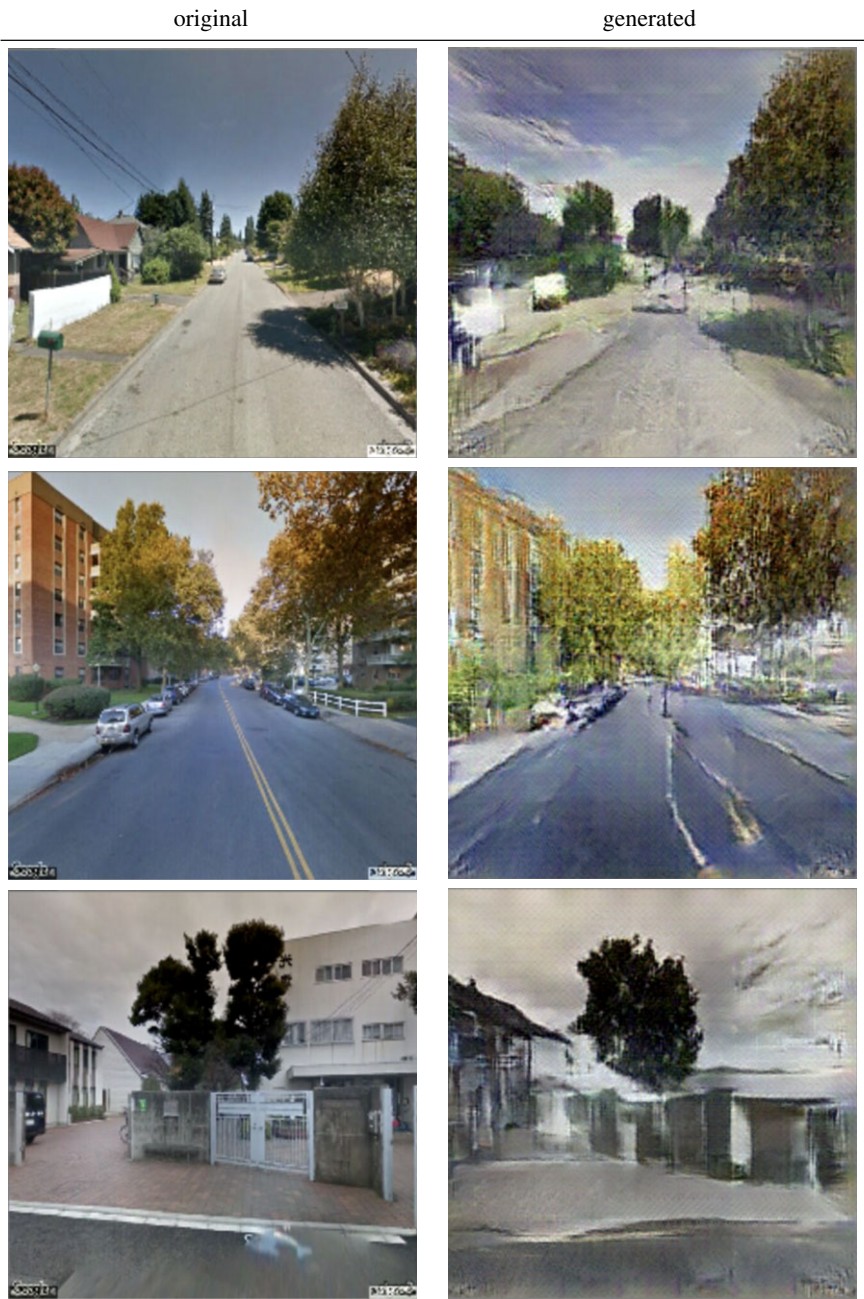

feature vector corresponding to $I_i$ and, as such, the resulting maximized $G(f)$ is $I_i$'s version 'morphed to become more beautiful'. Some examples of this morphing process can be seen in table 4.

The input image is also key. It makes little sense to beautify an already beautiful image, not least because such beautification process would result in a saturated template $\hat{I}_j$. For this reason, to generate an image that maximizes the beauty neuron in the classifier $C$, we restrict the input set to ugly scenes. We do the opposite when maximizing the ugly neuron.

### 3.4. Step 4: Returning a realistic beautified scene

We now have template scene $\hat{I}_j$ (which is a synthetic beautified version of original scene $I_i$) and need to retrieve a realistic-looking version of it. We do so by: (i) representing each of the scenes in our augmented set plus the synthetic image $\hat{I}_j$ as a 4096-dimensional feature vector derived from the FC7 layer of the PlacesNet [33], (ii) computing the Euclidean distance (as $L_2$ Norm) between $\hat{I}_j$'s feature vector and

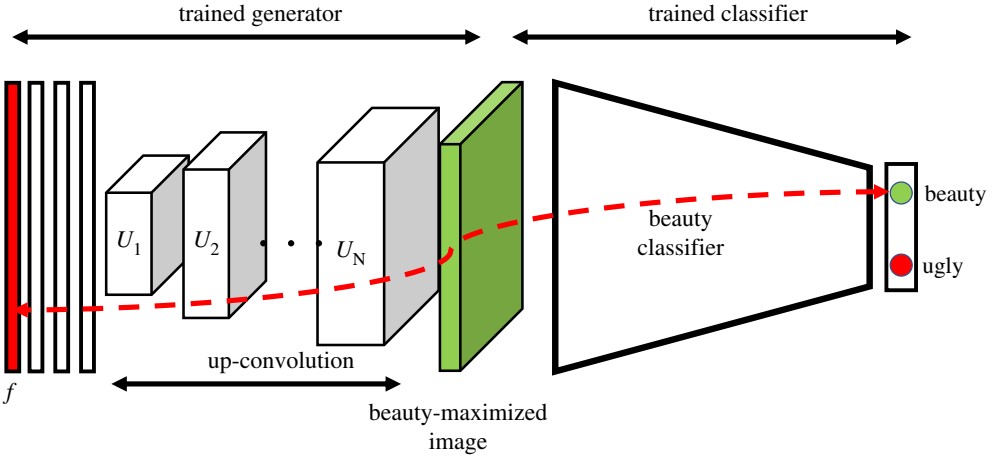

**Figure 5.** Architecture of the synthetic beauty generator. This consists of a generator of synthetic scenes concatenated with a beauty classifier. The green block is the beauty-maximized template $\hat{I}_j$, which is subject to forward and backward passes (red arrow) when optimizing for beauty.

each original scene's feature vector, and (iii) selecting the scene in our augmented set most similar (smaller distance) to $\hat{I}_j$. This results into the selection of the beautified scene $I_j$.

## 3.5. Step 5: Identifying characterizing urban elements

Since original scene $I_i$ and beautified scene $I_j$ are real scenes with the same structural characteristics (e.g. point of view, layout), we can easily compare them in terms of presence or absence of urban elements extracted by computer vision tools such as SegNet and PlacesNet. That is, we can determine how the original scene and its beautified version differ in terms of urban design elements.

# 4. Evaluation

The goal of FaceLift is to transform existing urban scenes into versions that: (i) people perceive more beautiful, (ii) contain urban elements typical of great urban spaces, (iii) are easy to interpret, and (iv) architects and urban planners find useful. To ascertain whether FaceLift meets that composite goal, we answer the following questions next:

**Q1** Do individuals perceive 'FaceLifted' scenes to be beautiful?
**Q2** Does our framework produce scenes that possess urban elements typical of great spaces?
**Q3** Which urban elements are mostly associated with beautiful scenes?
**Q4** Do architects and urban planners find FaceLift useful?

## 4.1. Q1 People's perceptions of beautified scenes

To ascertain whether 'FaceLifted' scenes are perceived by individuals as they are supposed to, we ran a crowd-sourcing experiment on Amazon Mechanical Turk. We randomly select 200 scenes, 100 beautiful and 100 ugly (taken at the bottom 10 and top 10 percentiles of the Trueskill's score distribution of figure 2). Our framework then transforms each ugly scene into its beautified version, and each beautiful scene into its corresponding 'uglified' version. These scenes are arranged into pairs, each of which contains the original scene and its beautified or uglified version. On Mechanical Turk, we only select verified masters as our crowd-sourcing workers (those with an approval rate above 90% during the past 30 days), pay them $0.1 per task, and ask each of them to choose the most beautiful scene for each given pair. We make sure to have at least three votes for each scene pair. Overall, our workers end up selecting the scenes that are actually beautiful 77.5% of the times, suggesting that 'FaceLifted' scenes are indeed perceived to be more beautiful by people.

## 4.2. Q2 Are beautified scenes great urban spaces?

To answer that question, we need to understand what makes a space great. After reviewing the literature in urban planning, we identify four factors associated with great places [1,37] (table 5): they mainly tend to be walkable, offer greenery, feel cosy, and be visually rich.

**Table 4.** Examples of the 'FaceLifting' process, which tends to add greenery, narrow roads and pavements.

| original ($I_i$) | latent beauty representation ($\hat{I}_j$) | beautified ($I_j$) |
| --- | --- | --- |

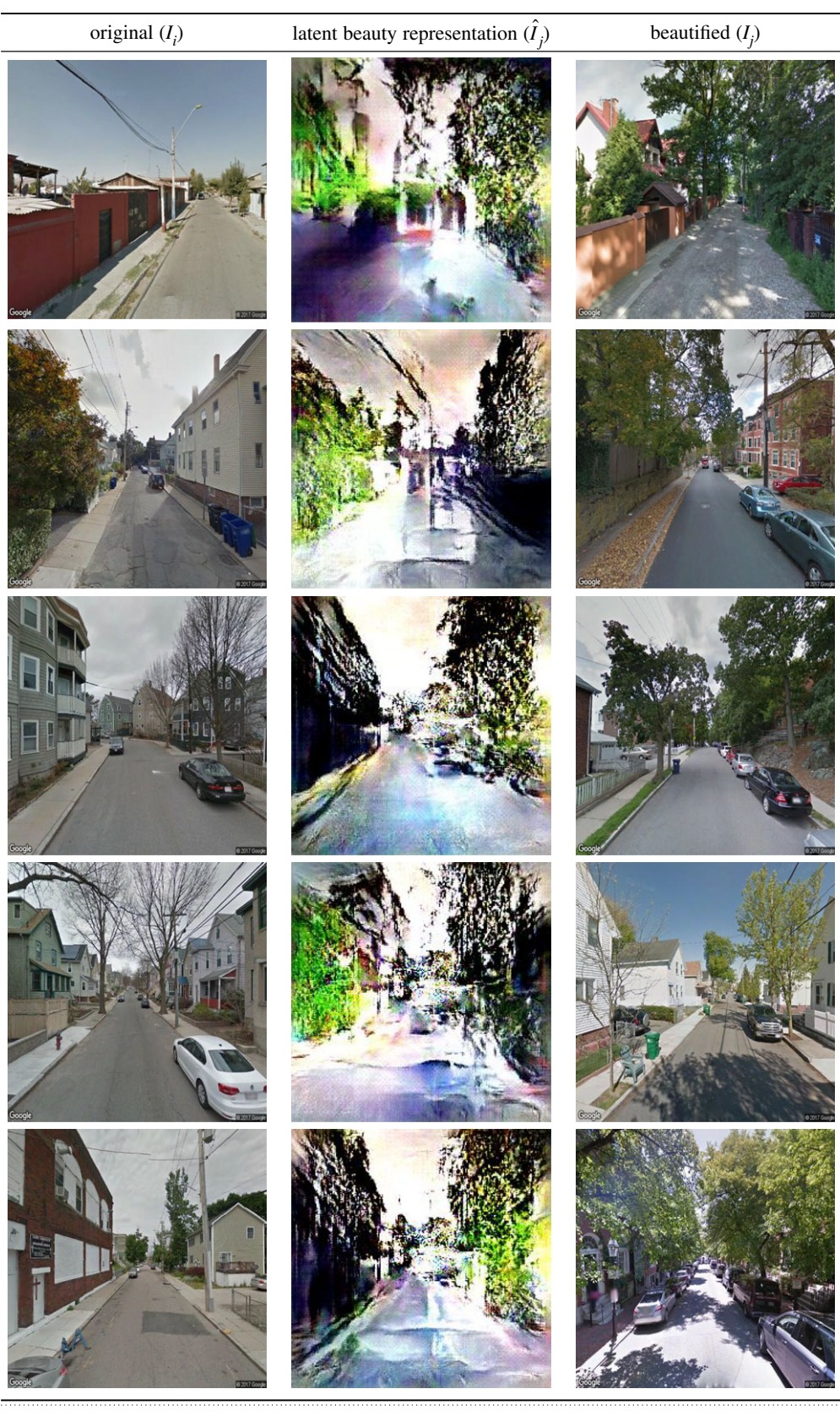

To automatically extract visual cues related to these four factors, we select 500 ugly scenes and 500 beautiful ones at random, transform them into their opposite aesthetic qualities (i.e. the ugly ones are beautified, and the beautiful ones are 'uglified'), and compare which urban elements related to the four factors distinguish uglified scenes from beautified ones.

**Table 5.** Urban design metrics.

| metric | description |
| --- | --- |
| walkability | walkable streets support people's natural tendency to explore spaces [37–39] |
| green spaces | the presence of greenery has repeatedly been found to impact people's well-being [1]. Under certain conditions, it could also promote social interactions [2]. Not all types of greenery have to be considered the same though: dense forests or unkempt greens might well have a negative impact [4] |
| landmarks | feeling lost is not a pleasant experience, and the presence of landmarks have been shown to contribute to the legibility and navigability of spaces [2,37,40,41] |
| privacy–openness | the sense of privacy conveyed by a place's structure (as opposed to a sense of openness) impacts its perception [37] |
| visual complexity | visual complexity is a measure of how diverse an urban scene is in terms of design materials, textures and objects [37]. The relationship between complexity and preferences generally follows an 'inverted-U' shape: we prefer places of medium complexity rather than places of low or high complexity [12] |

We extract labels from each of our 1000 scenes using two image classifiers. First, using PlacesNet [33], we label each of our scenes according to a classification containing 205 labels (reflecting, for example, landmarks, natural elements), and retain the five labels with highest confidence scores for the scene. We used a list of eight properties of walkable streets defined in previous work [42] as a guide to manually select only the PlacesNet labels that are related to walkability. These labels include, for example: *abbey, plaza, courtyard, garden, picnic area* and *park* (table 6 contains the exhaustive list). Second, using Segnet [43], we label each of our scenes according to a classification containing 12 labels. That is because Segnet is trained on dash-cam images, and classifies each scene pixel with one of these 12 labels: road, sky, trees, buildings, poles, signage, pedestrians, vehicles, bicycles, pavement, fences and road markings.

Having these two ways of labelling scenes, we can now test whether the expectations set by the literature of what makes urban spaces great (table 5) are met in the FaceLifted scenes.

### 4.2.1. *H1* Beautified scenes tend to be walkable.

We manually select only the PlacesNet labels that are related to walkability. These labels include, for example, *abbey, plaza, courtyard, garden, picnic area* and *park*. To test hypothesis *H1*, we count the number of walkability-related labels found in beautified scenes as opposed to those found in uglified scenes (figure 6): the former contain twice as many walkability labels than the latter. We then determine which types of scenes are associated with beauty (figure 7). Unsurprisingly, beautified scenes tend to show gardens, yards and small paths. By contrast, uglified ones tend to show built environment features such as shop fronts and broad roads. It is worth noting that walkability often acts as an enabler for other desirable properties of urban space (e.g. its restorative capability), and this might be the ultimate reason why our measure of walkability correlates with beauty.

### 4.2.2. *H2* Beautified scenes tend to offer green spaces

We manually select only the PlacesNet labels that are related to greenery. These labels include, for example, *fields, pasture, forest, ocean* and *beach*. Then, in our 1000 scenes, to test hypothesis *H2*, we count the number of nature-related labels found in beautified scenes as opposed to those found in uglified scenes (figure 6): the former contain more than twice as many nature-related labels than the latter. To test this hypothesis further, we compute the fraction of 'tree' pixels (using SegNet's label 'tree') in beautified and uglified scenes, and find that beautification adds 32% of tree pixels, while uglification removes 17% of them.

### 4.2.3. *H3* Beautified scenes tend to feel private and 'cosy'

To test hypothesis *H3*, we count the fraction of pixels that Segnet labelled as 'sky' and show the results in a bin plot in figure 8*a*: the *x*-axis has six bins (each of which represents a given range of sky fraction), and

**Table 6.** Classification of the PlacesNet labels into the four categories.

| architectural | walkable | landmark | natural |
|---|---|---|---|
| apartment building | abbey | airport | badlands |
| building facade | alley | amphitheatre | bamboo forest |
| construction site | boardwalk | amusement park | canyon |
| courthouse | botanical garden | arch | coast |
| driveway | corridor | baseball field | corn field |
| doorway | cottage garden | baseball stadium | creek |
| forest road | courtyard | basilica | desert (sand) |
| garbage dump | crosswalk | bridge | desert (vegetation) |
| golf course | fairway | castle | field (cultivated) |
| highway | food court | cathedral | field (wild) |
| hotel | forest path | cemetery | mountain |
| ice skating rink | formal garden | church | ocean |
| inn | herb garden | dam | orchard |
| motel | nursery | dock | pond |
| office building | outdoor market | fire station | rainforest |
| outdoor swimming pool | patio | football stadium | rice paddy |
| parking lot | pavilion | fountain | river |
| railroad track | picnic area | gas station | rock arch |
| residential neighbourhood | playground | harbour | sand bar |
| restaurant | plaza | hospital | sea cliff |
| runway | shopfront | lighthouse | ski slope |
| school house | topiary garden | mansion | sky |
| skyscraper | tree farm | mausoleum | snow field |
| slum | vegetable garden | pagoda | snowy mountain |
| supermarket | veranda | palace | swamp |
| tower | yard | racecourse | valley |
| water tower | | rope bridge | wheat field |
| wind farm | | ruin | |
| | | ski resort | |
| | | subway station | |
| | | train station | |
| | | temple | |
| | | wind mill | |

the $y$-axis shows the percentage of beautified versus uglified scenes that fall into each bin. Beautified scenes tend to be cosier (lower sky presence) than the corresponding original scenes.

### 4.2.4. H4 Beautified scenes tend to be visually rich

To quantify to which extent scenes are visually rich, we measure their visual complexity [37] as the amount of disorder in terms of distribution of (Segnet) urban elements in the scene

$$H_I = -\sum p(i) \log p(i), \tag{4.1}$$

where $i$ is the $i$th Segnet's label, and $p(i)$ is the proportion of urban scene $I$ containing the $i$th element. The total number of labels is 12. The higher $H_I$, the higher the scene's entropy, that is, the higher the scene's

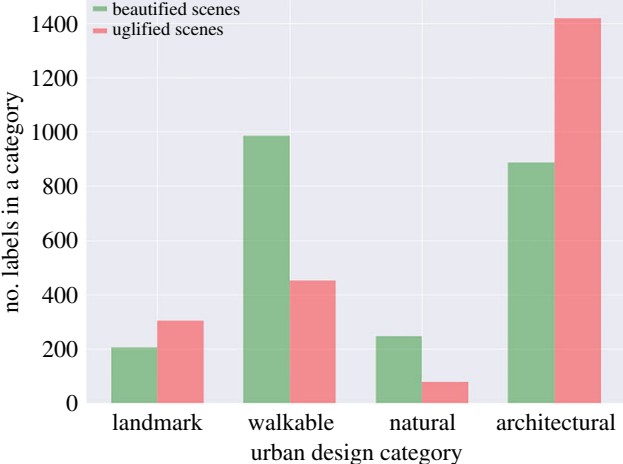

**Figure 6.** Number of labels in specific urban design categories (on the *x*-axis) found in beautified scenes as opposed to those found in uglified scenes.

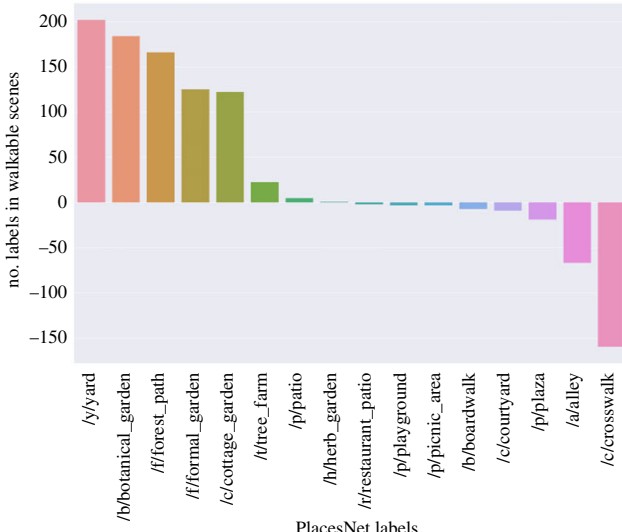

**Figure 7.** Count of specific walkability-related labels (on the *x*-axis) found in beautified scenes minus the count of the same labels found in uglified scenes.

complexity. It has been suggested that the relationship between complexity and pleasantness follows an 'inverted U' shape [12]: we prefer places of medium complexity rather than places of low or high complexity. To test that, we show the percentage of beautified scenes that fall into each complexity bin (figure 8*b*): we do not find a strong evidence of the 'inverted U' shape hypothesis, in that, beautified scenes are of low to medium complexity, while uglified ones are of high complexity.

## 4.3. Q3 Urban elements characterizing beautified scenes

To determine which urban elements are the best predictors of urban beauty and the extent to which they are so, we run a logistic regression, and, to ease interpretation, we do so on one pair of predictors at the time

$$\text{Pr(beautiful)} = \text{logit}^{-1}(\alpha + \beta_1 * V_1 + \beta_2 * V_2 + \beta_3 * V_1 . V_2), \tag{4.2}$$

where $V_1$ is the fraction of the scene's pixels marked with one Segnet's label, say, 'buildings' (over the total number of pixels), and $V_2$ is the fraction of the scene's pixels marked with another label, say, 'trees'. The result consists of three beta coefficients: $\beta_1$ reflects $V_1$'s contribution in predicting beauty, $\beta_2$ reflects $V_2$'s contribution, and $\beta_3$ is the interaction effect, that is, it reflects the contribution of the dependency between $V_1$ and $V_2$ in predicting beauty. We run logistic regressions on the five factors that have been found to be most predictive of urban beauty [1,2,37], and show the results in table 7.

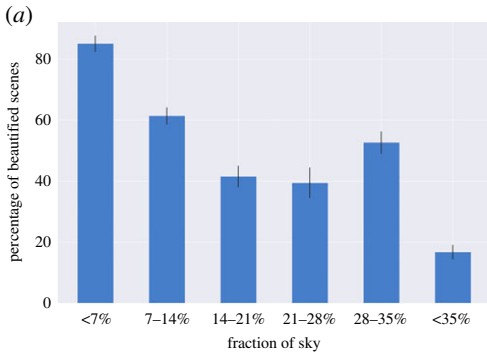
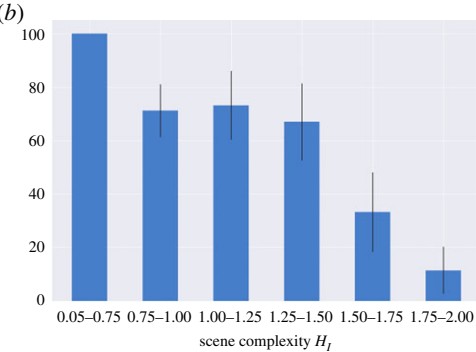

**Figure 8.** The percentage of beautified scenes (y-axis): (a) having an increasing presence of sky (on the x-axis) and (b) having an increasing level of visual richness (on the x-axis). The error bars represent standard errors obtained by random resampling of the data for 500 iterations.

**Table 7.** Coefficients of logistic regressions run on one pair of predictors at the time.

| pair of urban elements | $\beta_1$ | $\beta_2$ | $\beta_3$ | error rate (percentage) |
|---|---|---|---|---|
| buildings–trees | −0.032 | 0.084 | 0.005 | 12.7 |
| sky–buildings | −0.08 | −0.11 | 0.064 | 14.4 |
| roads–vehicles | −0.015 | −0.05 | 0.023 | 40.6 |
| sky–trees | 0.03 | 0.11 | −0.012 | 12.8 |
| roads–trees | 0.04 | 0.10 | −0.031 | 13.5 |
| roads–buildings | −0.05 | −0.097 | 0.04 | 20.2 |

Since we are using logistic regressions, the quantitative interpretation of the beta coefficients is eased by the 'divide by 4 rule' [44]: we can take the $\beta$ coefficients and 'divide them by 4 to get an upper bound of the predictive difference corresponding to a unit difference' in beauty [44]. For example, take the results in the first row of table 7. In the model $\Pr(\text{beautiful}) = \text{logit}^{-1}(\alpha - 0.032 \cdot \text{buildings} + 0.084 \cdot \text{trees} + 0.005 \cdot \text{buildings} \cdot \text{trees})$, we can divide −0.032/4 to get −0.008: a difference of 1% in the fraction of pixels being buildings corresponds to no more than a 0.8% *negative* difference in the probability of the scene being beautiful. In a similar way, a difference of 1% in the fraction of pixels being trees corresponds to no more than a 0.021% *positive* difference in the probability of the scene being beautiful. By considering the remaining results in table 7, we find that, across all pairwise comparisons, trees is the most positive element associated with beauty, while roads and buildings are the most negative ones. These results match previous literature on what makes urban design of spaces great [1,2,4,38,40], adding further external validity to our framework's beautification.

## 4.4. Q4 Do architects and urban planners find it useful?

To ascertain whether practitioners find FaceLift potentially useful, we build an interactive map of the city of Boston in which, for selected points, we show pairs of urban scenes before/after beautification (figure 9). We then send that map along with a survey to 20 experts in architecture, urban planning, and data visualization around the world. Questions were asked with a non-neutral response Likert scale (table 8). That is because previous work [45,46] has shown that such a scale: (i) pushes respondents to 'take a stance', given the absence of a neutral response, and (ii) works best if respondents are experts in the subject matter of the survey as responses of the 'I don't know' type tend to be rare (as has indeed been the case for our survey). The experts had to complete tasks in which they rated FaceLift based on how well it supports decision making, participatory urbanism, and the promotion of green spaces. According to our experts (table 8), the tool can very probably support decision making, probably support participatory urbanism and definitely promote green spaces. These results are also qualitatively supported by our experts' comments, which include: 'The maps reveal patterns that might not otherwise be apparent', 'The tool helps focusing on parameters to

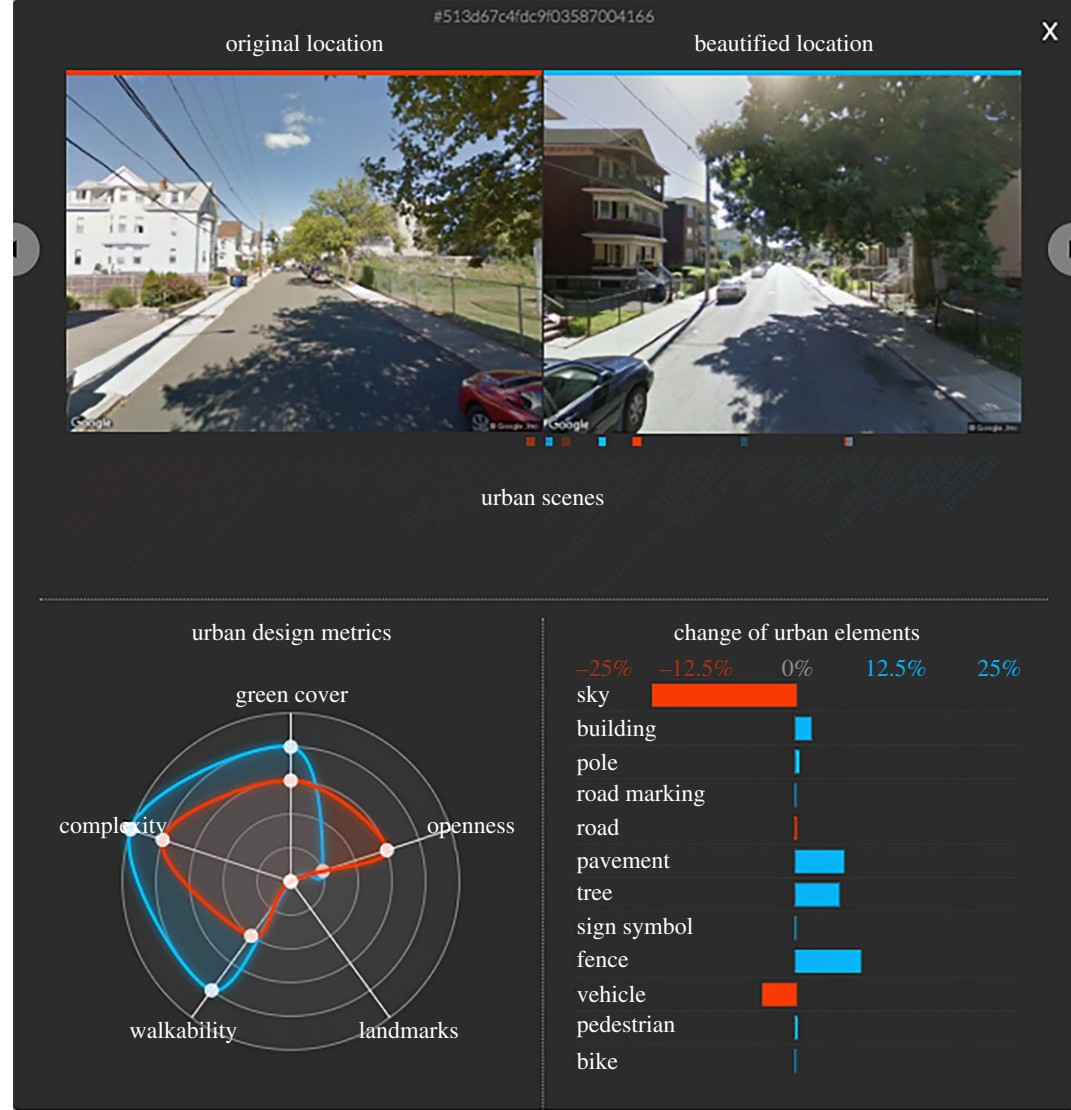

**Figure 9.** Interactive map of FaceLifted scenes in Boston.

**Table 8.** Urban experts polled about the extent to which an interactive map of 'FaceLifted' scenes promotes: (*a*) decision making; (*b*) citizen participation in urban planning and (*c*) promotion of green cities.

| use case | definitely not | probably not | probably | very probably | definitely |
|---|---|---|---|---|---|
| decision making | 4.8% | 9.5% | 38% | 28.6% | 19% |
| participatory urban planning | 0% | 4.8% | 52.4% | 23.8% | 19% |
| promote green cities | 4.8% | 0% | 47.6% | 19% | 28.6% |

identify beauty in the city while exploring it', and 'The metrics are nice. It made me think more about beautiful places needing a combination of criteria, rather than a high score on one or two dimensions. It made me realize that these criteria are probably spatially correlated.'

## 5. Discussion

FaceLift is a framework that automatically beautifies urban scenes by combining recent approaches of GANs and deep convolutional networks. To make it usable by practitioners, the framework is also able to explain which urban elements have been added/removed during the beautification process.

## 5.1. Limitations

FaceLift still faces some important challenges. The main limitation is that generative image models are still hard to control, especially when dealing with complex scenes containing multiple elements. Some of the beautifications suggested by our tool modify the scenes too dramatically to use them as blueprints for urban interventions (e.g. shifting buildings or broadening roads). This undesired effect is compounded by the restricted size and potential biases of the data that we use both for training and for selecting the scene most similar to the machine-generated image—which might result, for example, in generated scenes that are set in seasons or weather conditions that differ from the input image. To address these limitations, more work has to go into offering principled ways of fine-tuning the generative process, as well as into collecting reliable ground truth data on human perceptions. These data should ideally be stratified according to the people's characteristics that impact their perceptions. Performance assessment frameworks for the built environment (like the Living Building Challenge[3]) could provide a good source of non-traditional qualitative measures useful for training and validating the FaceLift algorithm. FaceLift still faces some important challenges. The main limitation is that generative image models are still hard to control, especially when dealing with complex scenes containing multiple elements. Some of the beautifications suggested by our tool modify the scenes too dramatically to use them as blueprints for urban interventions (e.g. shifting buildings or broadening roads). This undesired effect is compounded by the restricted size and potential biases of the data that we use both for training and for selecting the scene most similar to the machine-generated image—which might result, for example, in generated scenes that are set in seasons or weather conditions that differ from the input image. To address these limitations, more work has to go into offering principled ways of fine-tuning the generative process, as well as into collecting reliable ground truth data on human perceptions. This data should ideally be stratified according to the people's characteristics that impact their perceptions. Performance assessment frameworks for the built environment (like the Living Building Challenge[4]) could provide a good source of non-traditional qualitative measures useful for training and validating the FaceLift algorithm.

Another important limitation has to do with the complexity of the notion of beauty. There exists a wide spectrum of perceptive measures by which urban scenes could be considered beautiful. This is because the 'essence' of a place is socio-cultural and time-specific [47]. The collective perception of the urban environment evolves over time as its appearance and function change [48] as a result of shifting cultures, new urban policies, and placemaking initiatives [49]. An undiscerning, mechanistic application of machine learning tools to urban beautification might be undesirable because current technology cannot take into account most of these crucial aspects. FaceLift is no exception, and this is why we envision its use as a way to support new forms of placemaking rather than as a tool to replace traditional approaches. Nevertheless, we emphasize the need of a critical reflection on the implications of deploying such a technology, even when just in support of placemaking activities. In particular, it would be beneficial to study the impact of the transformative effect of FaceLift-inspired interventions on the ecosystem of the city [50,51] as well as exploring the need to pair its usage with practices and principles that might reduce any potential undesired side effect [52]

## 6. Conclusion

Despite these limitations, FaceLift has the potential to support urban interventions in scalable and replicable ways: it can be applied to an entire city (scalable), across a variety of cities (replicable).

We conceived FaceLift not as a technology to *replace* the decision-making process of planners and architects, but rather as a tool to *support* their work. FaceLift could aid the creative process of beautification of a city by suggesting imagined versions of what urban spaces could become after applying certain sets of interventions. We do not expect machine-generated scenes to equal the quality of designs done by experts. However, unlike the work of an expert, FaceLift is able to generate beautified scenes very fast (in seconds) and at scale (for an entire city), while quickly providing a numerical estimate of how much some urban elements should change to increase beauty. The user study we conducted suggests that these features make it possible to inspire the work of decision-makers and to nudge them into considering alternative approaches to urban interventions that might not otherwise be apparent. We believe this source of inspiration could benefit non-experts too, for

[3]See https://living-future.org/lbc/beauty-petal.

[4]See https://living-future.org/lbc/beauty-petal.

example, by helping residents to imagine a possible future for their cities and by motivating citizen action in the deployment of micro-interventions.

To turn existing spaces into something more beautiful, that will still be the duty of architecture. Yet, with technologies similar to FaceLift more readily available, the complex job of recreating restorative spaces in an increasingly urbanized world will be greatly simplified.

Data accessibility. The aggregated version of the data, the best performing models and all the supporting code can be accessed freely from http://goodcitylife.org/data.php.
Authors' contributions. S.J. carried out the design and development of the framework, did the data analysis, generated results and contributed to the writing of the article; D.Q., M.R. and L.M.A. conceived the study, provided insights and valuable direction to the analysis and contributed to the writing of the article; T.K. developed the data visualization for the results and conducted the expert survey; N.S. gave feedback on the manuscript. All authors gave final approval for publication.
Competing interests. We declare we have no competing interests.
Funding. We received no funding for this study.

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
