## [Reviewer comments · Royal Society Open Science]

Review History

RSOS-190987.R0 (Original submission)

Review form: Reviewer 1

Is the manuscript scientifically sound in its present form?

Yes

Are the interpretations and conclusions justified by the results?

Yes

Is the language acceptable?

Yes

Do you have any ethical concerns with this paper?

No

Have you any concerns about statistical analyses in this paper?

No

Recommendation?

Major revision is needed (please make suggestions in comments)

Comments to the Author(s)

The authors present a valuable study to help further understand what characteristics make a place more beautiful. I find the approach clever, as this is something that has been very difficult to quantify previously. The method for making sure the GAN output eventually results in realistic images is also very useful.

I do however have some concerns about the conclusion, specifically the line "We find that, across all the five metrics, the beautified scenes meet the expectations set by the literature on what great spaces tend to be made of". The two metrics I find less convincing are "Walkability" and "Privacy-Openness".

First of all, is there a full list of all the keywords used for each metric? This would be a useful addition to a Supplementary Information section. For "Walkability", I am concerned that the categories chosen (e.g. plaza, courtyard, park) tend to be places that people go to rest rather than walk. So I am undertrain that these categories are measuring something akin to walkability of a scene – they might be measuring something else entirely. This might still be a good urban design metric, but I am just not convinced it is measuring walkability.

As for "Privacy-Openness" – I am not convinced by the approach used to measure openness of a scene. Where a scene with tall trees would feel cozy, a scene with tall skyscrapers is likely to feel claustrophobic rather than cozy. So I am not convinced that lower sky presence equates to coziness, as this really depends on context. It is possible to extract Scene UNderstanding (SUN) Scene Attributes from Places205. These include elements such as far-away horizon, nohorizon, open area; perhaps this could be a helpful way of measuring the "Privacy-Openness" aspect of a scene.

I also note some small changes:

- How is similarity calculated on page 4? Is this cosine similarity?
- It would be useful to add citations to support p.14 "previous literature"

Review form: Reviewer 2 (Marcus Foth)

Is the manuscript scientifically sound in its present form?

Yes

Are the interpretations and conclusions justified by the results?

Yes

Is the language acceptable?

Yes

Do you have any ethical concerns with this paper?

No

Have you any concerns about statistical analyses in this paper?

No

Recommendation?

Accept with minor revision (please list in comments)

Comments to the Author(s)

This paper reports on a study that uses a deep learning approach to acquire the skill of "beautifying" urban scenes and identifying which elements were amended to achieve the result.

The paper is timely and significant considering the increased interest in urban informatics and urban science approaches. The paper is well written and clearly structured. I offer the following suggestions to further improve and strengthen the paper:

1. The paper specifies and explains key terms such as urban design, deep learning, and generative models, but not urban informatics. Perhaps add a definition and reference.
2. It would be useful for the reader to better understand the Place Pulse dataset, how it was created, and who participated in the curation and assignment of labels. What biases are present in this dataset? Has this been analysed?
3. Looking at the representative examples in Table 4, the "mechanism" or trained logic of the algorithm seems quite straightforward, as acknowledged by the authors, e.g. "adding greenery, narrow roads, and pavements." However, I also note that images in rows 3 and 4 shift from a winter scene (trees with no leaves) to a summer scene, or a grey sky to a blue sky. Further, some suggested beautifications shift entire structures such as buildings. These observations could be discussed, and then used to talk about two points: (a) limitations of the approach, and; (b) usefulness of the results (beyond what is covered in Q4).
4. Section Q4 did not convince me. The Likert scale sought to evaluate how well FaceLift supports decision making. It is not clear to me what is meant by decision making here. Similarly, I do not see how the substitution of an act of human creativity through a deep learning algorithm can be rated as "participatory urbanism" when there is nobody participating other than the machine.
5. This brings me to suggest the addition of a critical reflection and limitations section. Two examples of points that could be explored here:
 - (a) The mechanistic/positivist way the algorithm beautifies urban scenes risks becoming a cookie cutter as it does not take into account the full spectrum of authentic ways urban scenes can be activated and then perceived as beautiful. Similarly to how a leafless tree in winter is perceived less beautiful than a lush, leafy tree in summer, there are influences of people, urban policies, placemaking initiatives that impact on the notion of "beauty." Norberg-Schulz (1980) uses a phenomenology approach to describe the "essence" of a place, which is socio-culturally and time-specific. Brand (1997) traces the development of a street scene / building façade over time as it changes through renovations, modifications, and customisations and as a result, perceptions change. In my own work (2017), I reviewed placemaking interventions and explored participatory forms of citymaking.

Norberg-Schulz, C. (1980). *Genius loci: Towards a phenomenology of architecture*. New York, NY: Rizzoli.

Brand, S. (1997). *How Buildings Learn: What Happens After They're Built (Rev.)*. London: Phoenix Illustrated.

Foth, M. (2017). Lessons from Urban Guerrilla Placemaking for Smart City Commons. In Proceedings of the 8th International Conference on Communities and Technologies (C&T '17). ACM, New York, NY, USA, 32-35. DOI: <https://doi.org/10.1145/3083671.3083707>

(b) The positivist paradigm of urban science has been critiqued for its technocratic worldview, and the FaceLift study would benefit from a critical reflection by the authors that picks up on some of these points, e.g.:

Kitchin, R. (2017). Thinking critically about and researching algorithms. *Information, Communication and Society*, 20(1), 14-29. <https://doi.org/10.1080/1369118X.2016.1154087>

Dourish, P. (2016). Algorithms and their others: Algorithmic culture in context. *Big Data & Society*, 3(2). <https://doi.org/10.1177/2053951716665128>

Kitchin, R. (2016). The ethics of smart cities and urban science. *Philosophical Transactions. Series A, Mathematical, Physical, and Engineering Sciences*, 374(2083). <https://doi.org/10.1098/rsta.2016.0115>

6. A suggestion for future work: The Living Building Challenge (LBC) is a performance assessment framework for the built environment that introduces non-traditional and qualitative measures such as beauty. Those buildings and architectural projects that have been assessed by the LBC could perhaps offer a complementary dataset for additional ground truthing from another perspective: <https://living-future.org/lbc/beauty-petal/>

Decision letter (RSOS-190987.R0)

12-Sep-2019

Dear Mr Joglekar,

The editors assigned to your paper ("FaceLift: A transparent deep learning framework to beautify urban scenes") have now received comments from reviewers. We would like you to revise your paper in accordance with the referee and Associate Editor suggestions which can be found below (not including confidential reports to the Editor). Please note this decision does not guarantee eventual acceptance.

Please submit a copy of your revised paper before 05-Oct-2019. Please note that the revision deadline will expire at 00.00am on this date. If we do not hear from you within this time then it will be assumed that the paper has been withdrawn. In exceptional circumstances, extensions may be possible if agreed with the Editorial Office in advance. We do not allow multiple rounds of revision so we urge you to make every effort to fully address all of the comments at this stage. If deemed necessary by the Editors, your manuscript will be sent back to one or more of the original reviewers for assessment. If the original reviewers are not available, we may invite new reviewers.

To revise your manuscript, log into <http://mc.manuscriptcentral.com/rsos> and enter your Author Centre, where you will find your manuscript title listed under "Manuscripts with Decisions." Under "Actions," click on "Create a Revision." Your manuscript number has been

appended to denote a revision. Revise your manuscript and upload a new version through your Author Centre.

- Data accessibility

If you wish to submit your supporting data or code to Dryad (<http://datadryad.org/>), or modify your current submission to dryad, please use the following link:
<http://datadryad.org/submit?journalID=RSOS&manu=RSOS-190987>

- Competing interests

- Authors' contributions

- Acknowledgements

- Funding statement

on behalf of Dr Danica Vukadinovic Greetham (Associate Editor) and Marta Kwiatkowska (Subject Editor)
openscience@royalsociety.org

Comments to Author:

Reviewers' Comments to Author:

Reviewer: 1

Comments to the Author(s)

The authors present a valuable study to help further understand what characteristics make a place more beautiful. I find the approach clever, as this is something that has been very difficult to quantify previously. The method for making sure the GAN output eventually results in realistic images is also very useful.

I do however have some concerns about the conclusion, specifically the line "We find that, across all the five metrics, the beautified scenes meet the expectations set by the literature on what great spaces tend to be made of". The two metrics I find less convincing are "Walkability" and "Privacy-Openness".

First of all, is there a full list of all the keywords used for each metric? This would be a useful addition to a Supplementary Information section. For "Walkability", I am concerned that the categories chosen (e.g. plaza, courtyard, park) tend to be places that people go to rest rather than walk. So I am undertrain that these categories are measuring something akin to walkability of a scene - they might be measuring something else entirely. This might still be a good urban design metric, but I am just not convinced it is measuring walkability.

As for "Privacy-Openness" - I am not convinced by the approach used to measure openness of a scene. Where a scene with tall trees would feel cozy, a scene with tall skyscrapers is likely to feel claustrophobic rather than cozy. So I am not convinced that lower sky presence equates to coziness, as this really depends on context. It is possible to extract Scene UNDERstanding (SUN) Scene Attributes from Places205. These include elements such as far-away horizon, nohorizon,

open area; perhaps this could be a helpful way of measuring the "Privacy-Openness" aspect of a scene.

I also note some small changes:

- How is similarity calculated on page 4? Is this cosine similarity?
- It would be useful to add citations to support p.14 "previous literature"

Reviewer: 2

Comments to the Author(s)

This paper reports on a study that uses a deep learning approach to acquire the skill of "beautifying" urban scenes and identifying which elements were amended to achieve the result.

The paper is timely and significant considering the increased interest in urban informatics and urban science approaches. The paper is well written and clearly structured. I offer the following suggestions to further improve and strengthen the paper:

1. The paper specifies and explains key terms such as urban design, deep learning, and generative models, but not urban informatics. Perhaps add a definition and reference.

2. It would be useful for the reader to better understand the Place Pulse dataset, how it was created, and who participated in the curation and assignment of labels. What biases are present in this dataset? Has this been analysed?

3. Looking at the representative examples in Table 4, the "mechanism" or trained logic of the algorithm seems quite straightforward, as acknowledged by the authors, e.g. "adding greenery, narrow roads, and pavements." However, I also note that images in rows 3 and 4 shift from a winter scene (trees with no leaves) to a summer scene, or a grey sky to a blue sky. Further, some suggested beautifications shift entire structures such as buildings. These observations could be discussed, and then used to talk about two points: (a) limitations of the approach, and; (b) usefulness of the results (beyond what is covered in Q4).

4. Section Q4 did not convince me. The Likert scale sought to evaluate how well FaceLift supports decision making. It is not clear to me what is meant by decision making here. Similarly, I do not see how the substitution of an act of human creativity through a deep learning algorithm can be rated as "participatory urbanism" when there is nobody participating other than the machine.

5. This brings me to suggest the addition of a critical reflection and limitations section. Two examples of points that could be explored here:

(a) The mechanistic/positivist way the algorithm beautifies urban scenes risks becoming a cookie cutter as it does not take into account the full spectrum of authentic ways urban scenes can be activated and then perceived as beautiful. Similarly to how a leafless tree in winter is perceived less beautiful than a lush, leafy tree in summer, there are influences of people, urban policies, placemaking initiatives that impact on the notion of "beauty." Norberg-Schulz (1980) uses a phenomenology approach to describe the "essence" of a place, which is socio-culturally and time-specific. Brand (1997) traces the development of a street scene / building façade over time as it changes through renovations, modifications, and customisations and as a result, perceptions change. In my own work (2017), I reviewed placemaking interventions and explored participatory forms of citymaking.

Norberg-Schulz, C. (1980). *Genius loci: Towards a phenomenology of architecture*. New York, NY: Rizzoli.

Brand, S. (1997). *How Buildings Learn: What Happens After They're Built (Rev.)*. London: Phoenix Illustrated.

Foth, M. (2017). Lessons from Urban Guerrilla Placemaking for Smart City Commons. In *Proceedings of the 8th International Conference on Communities and Technologies (C&T '17)*. ACM, New York, NY, USA, 32-35. DOI: <https://doi.org/10.1145/3083671.3083707>

(b) The positivist paradigm of urban science has been critiqued for its technocratic worldview, and the FaceLift study would benefit from a critical reflection by the authors that picks up on some of these points, e.g.:

Kitchin, R. (2017). Thinking critically about and researching algorithms. *Information, Communication and Society*, 20(1), 14-29. <https://doi.org/10.1080/1369118X.2016.1154087>

Dourish, P. (2016). Algorithms and their others: Algorithmic culture in context. *Big Data & Society*, 3(2). <https://doi.org/10.1177/2053951716665128>

Kitchin, R. (2016). The ethics of smart cities and urban science. *Philosophical Transactions. Series A, Mathematical, Physical, and Engineering Sciences*, 374(2083). <https://doi.org/10.1098/rsta.2016.0115>

6. A suggestion for future work: The Living Building Challenge (LBC) is a performance assessment framework for the built environment that introduces non-traditional and qualitative measures such as beauty. Those buildings and architectural projects that have been assessed by the LBC could perhaps offer a complementary dataset for additional ground truthing from another perspective: <https://living-future.org/lbc/beauty-petal/>

Author's Response to Decision Letter for (RSOS-190987.R0)

See Appendix A.

RSOS-190987.R1 (Revision)

Review form: Reviewer 1

Is the manuscript scientifically sound in its present form?

Yes

Are the interpretations and conclusions justified by the results?

Yes

Is the language acceptable?

Yes

Do you have any ethical concerns with this paper?

No

Have you any concerns about statistical analyses in this paper?

No

Recommendation?

Accept as is

Comments to the Author(s)

Thank you for making the requested changes. I still find the title for H3 "Beautified scenes tend to feel private and cozy" misleading as I don't think this phrase adequately represents the lack of sky in a picture and would strongly urge the author to consider a more intuitive title. This is what confused me on the first reading when I suggested measuring openness. Also, to be clear, perhaps on p.9 in "they mainly tend to be walkable, offer greenery, feel cozy, and be visually rich.", "feel cozy" should also be replaced with a phrase that better represents the actual measurement. I leave this up to the authors though as it is such a minor point and I am satisfied that while the phrase is misleading, the method of measurement is still explained very clearly.

Decision letter (RSOS-190987.R1)

28-Oct-2019

Dear Mr Joglekar,

I am pleased to inform you that your manuscript entitled "FaceLift: A transparent deep learning framework to beautify urban scenes" is now accepted for publication in Royal Society Open Science.

Kind regards,
Anita Kristiansen
Editorial Coordinator
Royal Society Open Science
openscience@royalsociety.org

on behalf of Dr Danica Vukadinovic Greetham (Associate Editor) and Marta Kwiatkowska (Subject Editor)
openscience@royalsociety.org

Associate Editor Comments to Author (Dr Danica Vukadinovic Greetham):

Comments to the Author:

Thank you for taking into account the reviewers' suggestions and making the requested changes.

Reviewer comments to Author:

Reviewer: 1

Comments to the Author(s)

Thank you for making the requested changes. I still find the title for H3 "Beautified scenes tend to feel private and cozy" misleading as I don't think this phrase adequately represents the lack of sky in a picture and would strongly urge the author to consider a more intuitive title. This is what confused me on the first reading when I suggested measuring openness. Also, to be clear, perhaps on p.9 in "they mainly tend to be walkable, offer greenery, feel cozy, and be visually rich.", "feel cozy" should also be replaced with a phrase that better represents the actual measurement. I leave this up to the authors though as it is such a minor point and I am satisfied that while the phrase is misleading, the method of measurement is still explained very clearly.

Appendix A

Response to the reviews for Royal Society Open Science submission “FaceLift: A transparent deep learning framework to beautify urban scenes”

Sagar Joglekar, Daniele Quercia, Miriam Redi, Luca Aiello, Tobias Kauer, Nishanth Sastry

We would like to express our sincere thanks to the Editors for their very detailed and constructive comments. We have worked to address all their concerns in the revised version of the manuscript.

Summary of reviewer requests

Reviewers made certain important points which we try to address in this revision:

- 1. Reviewer 1 expressed concern about the validity of the “Walkability” metric actually quantifying walkability of places*
- 2. Reviewer 1 expressed concern about the validity of the “Openness” metric actually quantifying open nature of places. They also suggested trying out Scene Understanding (SUN) attributes from places 205 to address this.*
- 3. Reviewer 2 recommended adding definition for urban informatics along with a reference.*
- 4. Reviewer 2 recommended discussion the PlacePulse dataset in more detail. “ how it was created, and who participated in the curation and assignment of labels. What biases are present in this dataset? Has this been analysed?”*
- 5. Reviewer 2: Looking at the representative examples in Table 4, the “mechanism” or trained logic of the algorithm seems quite straightforward, as acknowledged by the authors, e.g. “adding greenery, narrow roads, and pavements.” However, I also note that images in rows 3 and 4 shift from a winter scene (trees with no leaves) to a summer scene, or a grey sky to a blue sky. Further, some suggested beautifications shift entire structures such as buildings. These observations could be discussed, and then used to talk about two points: (a) limitations of the approach, and; (b) usefulness of the results (beyond what is covered in Q4).*
- 6. Reviewer 2: Section Q4 did not convince me. The Likert scale sought to evaluate how well FaceLift supports decision making. It is not clear to me what is meant by decision making here. Similarly, I do not see how the substitution of an act of human creativity through a deep learning algorithm can be rated as “participatory urbanism” when there is nobody participating other than the machine.*

7. Reviewer 2 recommended adding critical reflections about the framework and have kindly given pointers to do so.

We have thoroughly revised the paper to address all the comments from the Reviewers. Below, we provide detailed answers to each of their comments individually. In the revised manuscript, changes are highlighted in blue.

Requests from Reviewer 1

First of all, is there a full list of all the keywords used for each metric? This would be a useful addition to a Supplementary Information section. For "Walkability", I am concerned that the categories chosen (e.g. plaza, courtyard, park) tend to be places that people go to rest rather than walk. So I am undertrain(sic) that these categories are measuring something akin to walkability of a scene they might be measuring something else entirely. This might still be a good urban design metric, but I am just not convinced it is measuring walkability.

To identify the PlacesNet labels that belong to the Walkable category, we rely on 8 properties of walkable streets defined in previous work [11]: *Road safety, Easy to cross, Sidewalks, Hilliness, Navigation, Safety from crime, Smart and beautiful, Fun and relaxing*. We use these 8 properties as a guide to determine which PlacesNet labels could be classified as Walkable. We summarize this categorization in a new Table (see the bottom of this letter and Table 6 in the new manuscript). The urban elements in the Walkable category serve different functions, yet they are either walkable spaces or facilities that are most often situated in walkable areas. We do agree with the Reviewer that walkability acts often as an enabler for other desirable properties of space (e.g., its restorative potential). In the new version of the manuscript, in Section "Q2 Are beautified scenes great urban spaces?," we acknowledge that our walkability measure might correlate with the notion of beauty because it acts as a proxy for higher-order properties that are enabled by walkability:

“ We used a list of 8 properties of walkable streets defined in previous work [11] as a guide to manually select only the PlacesNet labels that are related to walkability. These labels include, for example: *abbey, plaza, courtyard, garden, picnic area, and park* (Table 6 contains the exhaustive list) [...] Unsurprisingly, beautified scenes tend to show gardens, yards, and small paths. By contrast, uglified ones tend to show built environment features such as shop fronts and broad roads. It is worth noting that walkability often acts as an enabler for other desirable properties of urban space (e.g., its restorative capability), and this might be the ultimate reason why our measure of walkability correlates with beauty. ”

As for "Privacy-Openness" I am not convinced by the approach used to measure openness of a scene. Where a scene with tall trees would feel cozy, a scene with tall skyscrapers is likely to feel claustrophobic rather than cozy. So I am not convinced that lower sky presence equates to coziness, as this really depends on context. It is possible to extract Scene UNDERstanding (SUN) Scene Attributes from Places205. These

include elements such as far-away horizon, nohorizon, open area; perhaps this could be a helpful way of measuring the "Privacy-Openness" aspect of a scene.

We counted the amount of "sky pixels" to measure openness because we wanted to characterize this property on a continuous spectrum, which is not possible with other frameworks. As we will discuss next, the Scene UNderstanding (SUN) [10, 12] framework does not have a direct way to quantify the amount of visible sky. However, we followed the Reviewer's suggestion and conducted an additional experiment using SUN.

We first extracted scene attributes from all the beautified and uglified scenes using SUN. Within each group, we counted the number of pictures with a given SUN attribute i . We denote these counts as ϕ_i^{beauty} and ϕ_i^{ugly} . We then calculated the differences of these counts between the two groups: $\Delta(\phi_i) = \phi_i^{beauty} - \phi_i^{ugly}$. Figure 1 in this letter shows the values of these differences across attributes. Positive (negative) values indicate that the attribute is more prevalent in the beautified (uglified) group. Scene attributes like *trees*, and *foliage* are found more frequently in beautified scenes, whereas attributes like *man-made*, *driving*, and *no horizon* are found predominantly in uglified scenes. These results partially confirm our findings, in that they reveal the broad distinction between beautiful environments with natural elements vs. places where only man-made elements are visible.

Categories like *open area*, *enclosed area*, and *no horizon* do not necessarily reflect the amount of visible sky, as it is shown in the discussion from the original paper [10]. For example, all pictures characterized by the *open areas* attribute depict outdoor places but might include either open-sky scenes or scenes where the sky is occluded by trees. Similarly, scenes with the *enclosed areas* attribute depict walled constructions that might be in areas with different levels of exposure to open sky.

In short, SUN provides some support to our findings but its results cannot provide a direct measure of openness. The Reviewer is right in noting that, in general, lack of openness could indicate cozyness as well as visual oppression. In our specific dataset, spaces with more visible sky tend, on average, to be less visually appealing.

We did not report these additional results in the paper, but if the Reviewer thinks it would be useful to have them in the camera-ready version, we will be happy to include them.

Also note some small changes:

- - *How is similarity calculated on page 4? Is this cosine similarity?*
- - *It would be useful to add citations to support p.14 "previous literature"*

We used Euclidean distance and we made sure to mention it in the revised version in page 4 (step 4). In page 14, we added 5 citations that cover the "previous literature" we refer to.

Requests from Reviewer 2

The paper specifies and explains key terms such as urban design, deep learning, and generative models, but not urban informatics. Perhaps add a definition and reference.

Fig. 1:

As suggested by the Reviewer, in the introduction we added a definition of “urban informatics” together with a couple of supporting references:

“Our work contributes to the field of urban informatics, an interdisciplinary area of research that studies practices and experiences across urban contexts and creates new digital tools to improve those experiences [4, 6].”

It would be useful for the reader to better understand the Place Pulse dataset, how it was created, and who participated in the curation and assignment of labels. What biases are present in this dataset? Has this been analysed? ...

We agree with the Reviewer that a more self-contained description of the Place Pulse dataset and of its

potential biases is in order. We added the following paragraph in the Section “Curating Urban Scenes” (page 4):

“ To begin with, we need highly curated training data with labels reflecting urban beauty. We start with the Place Pulse dataset that contains a set of 110k Google street view images from 56 major cities across 28 countries around the world [3]. The pictures were labeled by volunteers through an ad-hoc crowdsourcing website¹. Volunteers were shown random pairs of images and asked to select which scene looked more beautiful, safe, lively, boring, wealthy, and depressing. At the time of writing, 1.2 million pairwise comparisons were generated by 82k online volunteers from 162 countries, with a good mix of people residing in both developed and developing countries. To our knowledge, no independent systematic analysis of the biases of Place Pulse has been conducted yet. However, it is reasonable to expect that representation biases are minimized by the substantial size of the dataset, the wide variety of places represented, and the diversity of gender, racial, and cultural backgrounds of the raters. ”

Looking at the representative examples in Table 4, the “mechanism” or trained logic of the algorithm seems quite straightforward, as acknowledged by the authors, e.g. “adding greenery, narrow roads, and pavements.” However, I also note that images in rows 3 and 4 shift from a winter scene (trees with no leaves) to a summer scene, or a grey sky to a blue sky. Further, some suggested beautifications shift entire structures such as buildings. These observations could be discussed, and then used to talk about two points: (a) limitations of the approach, and; (b) usefulness of the results (beyond what is covered in Q4).

We agree with the Reviewer that the examples in Table 4 expose some of the limitations of our approach. These can be broadly summarized by saying that generative image models are still hard to control, especially when dealing with complex scenes with several elements. This shortcoming is compounded by the restricted size of training data. We briefly mentioned this limitation in the previous version of the paper; in the new manuscript, we expand on that point:

“ The main limitation is that generative image models are still hard to control, especially when dealing with complex scenes containing multiple elements. Some of the beautifications suggested by our tool modify the scenes too dramatically to use them as blueprints for urban interventions (e.g., shifting buildings or broadening roads). This undesired effect is compounded by the restricted size and potential biases of the data that we use both for training and for selecting the scene most similar to the machine-generated image—which might result, for example, in generated scenes that are set in seasons or weather conditions that differ from the input image. To address these limitations, more work has to go into offering principled ways of fine-tuning the generative process, as well as into collecting reliable ground truth data on human perceptions. This data should ideally be stratified according to the people’s characteristics that impact their perceptions. ”

Even though this limitation partly restricts the capacity of our tool, we still argue for its potential to simplify and democratize the process of creating restorative spaces, as we detail in the next reply.

¹ <http://pulse.media.mit.edu>

Section Q4 did not convince me. The Likert scale sought to evaluate how well FaceLift supports decision making. It is not clear to me what is meant by decision making here. Similarly, I do not see how the substitution of an act of human creativity through a deep learning algorithm can be rated as “participatory urbanism” when there is nobody participating other than the machine.

We thank the Reviewer for allowing us to clarify this point, as we realize we could be clearer on the purpose for which FaceLift is intended. We added the following discussion in the conclusions section, hoping that it will serve to clarify the intended use cases for our tool:

“ We conceived FaceLift not as a technology to *replace* the decision making process of planners and architects, but rather as a tool to *support* their work. FaceLift could aid the creative process of beautification of a city by suggesting imagined versions of what urban spaces could become after applying certain sets of interventions. We do not expect machine-generated scenes to equal the quality of designs done by experts. However, unlike the work of an expert, FaceLift is able to generate beautified scenes very fast (in seconds) and at scale (for an entire city), while quickly providing a numerical estimate of how much some urban elements should change to increase beauty. The user study we conducted suggests that these features make it possible to inspire the work of decision makers and to nudge them into considering alternative approaches to urban interventions that might not otherwise be apparent. We believe this source of inspiration could advantage non-experts too, for example by helping residents to imagine a possible future for their cities and by motivating citizen action in the deployment of micro-interventions. ”

This brings me to suggest the addition of a critical reflection and limitations section. Two examples of points that could be explored here:

(a) The mechanistic/positivist way the algorithm beautifies urban scenes risks becoming a cookie cutter as it does not take into account the full spectrum of authentic ways urban scenes can be activated and then perceived as beautiful. Similarly to how a leafless tree in winter is perceived less beautiful than a lush, leafy tree in summer, there are influences of people, urban policies, placemaking initiatives that impact on the notion of “beauty.” Norberg-Schulz (1980) uses a phenomenology approach to describe the “essence” of a place, which is socio-culturally and time-specific. Brand (1997) traces the development of a street scene / building faade over time as it changes through renovations, modifications, and customisations and as a result, perceptions change. In my own work (2017), I reviewed placemaking interventions and explored participatory forms of citymaking.

- *Norberg-Schulz, C. (1980). *Genius loci: Towards a phenomenology of architecture*. New York, NY: Rizzoli.*
- *Brand, S. (1997). *How Buildings Learn: What Happens After Theyre Built (Rev.)*. London: Phoenix Illustrated.*
- *Foth, M. (2017). *Lessons from Urban Guerrilla Placemaking for Smart City Commons*. In *Proceedings of the 8th International Conference on Communities and Technologies (C&T '17)*. ACM, New York, NY, USA, 32-35. DOI: <https://doi.org/10.1145/3083671.3083707>*

(b) *The positivist paradigm of urban science has been critiqued for its technocratic worldview, and the FaceLift study would benefit from a critical reflection by the authors that picks up on some of these points, e.g.:*

- *Kitchin, R. (2017). Thinking critically about and researching algorithms. Information, Communication and Society, 20(1), 1429. <https://doi.org/10.1080/1369118X.2016.1154087>*
- *Dourish, P. (2016). Algorithms and their others: Algorithmic culture in context. Big Data & Society, 3(2). <https://doi.org/10.1177/2053951716665128>*
- *Kitchin, R. (2016). The ethics of smart cities and urban science. Philosophical Transactions. Series A, Mathematical, Physical, and Engineering Sciences, 374(2083). <https://doi.org/10.1098/rsta.2016.0115>*

We fully agree with these remarks and with the need of emphasizing such limitations and potential risks. We took these suggestions onboard and further expanded our limitations section by following what the Reviewer exposed so expertly in their comment.

“ There exists a wide spectrum of perceptive measures by which urban scenes could be considered beautiful. This is because the “essence” of a place is socio-cultural and time-specific [9]. The collective perception of the urban environment evolves over time as its appearance and function change [1] as a result of shifting cultures, new urban policies, and placemaking initiatives [5]. An undiscerning, mechanistic application of machine learning tools to urban beautification might be undesirable because current technology cannot take into account most of these crucial aspects. FaceLift is no exception, and this is why we envision its use as a way to support new forms of placemaking rather than as a tool to replace traditional approaches. Nevertheless, we emphasize the need of a critical reflection on the implications of deploying such a technology, even when just in support of placemaking activities. In particular, it would be beneficial to study the impact of the transformative effect of FaceLift-inspired interventions on the ecosystem of the city [2, 8] as well as exploring the need to pair its usage with practices and principles that might reduce any potential undesired side effect [7] ”

A suggestion for future work: The Living Building Challenge (LBC) is a performance assessment framework for the built environment that introduces non-traditional and qualitative measures such as beauty. Those buildings and architectural projects that have been assessed by the LBC could perhaps offer a complementary dataset for additional ground truthing from another perspective: <https://living-future.org/lbc/beauty-petal/>

We thank the Reviewer for this relevant pointer. We added a mention to LBC as a possible source of validation data that is orthogonal to what we considered in this work.

We want to express our sincere thanks to the Editor and to the Reviewers for all their constructive feedback. We hope they will find the new version of the paper much improved.

Architectural	Walkable	Landmark	Natural
Apartment building	Abbey	Airport	Badlands
Building Facade	Alley	Amphitheatre	Bamboo Forest
Construction Site	Boardwalk	Amusement Park	Canyon
Courthouse	Botanical Garden	Arch	Coast
Drive way	Corridor	Amphitheatre	Corn field
Door way	Cottage garden	Baseball Field	Creek
Forest road	Courtyard	Basilica	Desert (Sand)
Garbage dump	Crosswalk	Bridge	Field (cultivated)
Golf course	Fairway	Castle	Field (wild)
Highway	Food court	Wind mill	Mountain
Hotel	Forest path	Cathedral	Snowy Mountain
Inn	Formal Garden	Church	Ocean
Ice skating rink	Herb Garden	Dam	Orchard
Motel	Outdoor Market	Dock	Pond
Office building	Nursery	Cemetery	Rainforest
Parking Lot	Patio	Fire station	Rice paddy
Railroad track	Pavilion	Fountain	River
Residential neighbourhood	Picnic area	Gas Station	Rock arch
Restaurant	Playground	Harbour	Sand bar
Runway	Plaza	Hospital	Sea Cliff
School House	Patio	Lighthouse	Ski slope
Skyscraper	Shopfront	Mansion	Sky
Slum	Topiary garden	Mausoleum	Snow field
Supermarket	Tree farm	Pagoda	Swamp
Outdoor swimming pool	Veranda	Palace	Valley
Tower	Vegetable garden	Racecourse	Wheat field
Water tower	Yard	Ruin	Desert (vegetation)
Wind farm		Rope Bridge	
		Ski Resort	
		Baseball stadium	
		Football stadium	
		Subway Station	
		Train Station	
		Temple	

Tab. 1: Classification of the PlacesNet labels into the four categories.

References

- [1] S. Brand. *How buildings learn: What happens after they're built*. Penguin, 1995.
- [2] P. Dourish. Algorithms and their others: Algorithmic culture in context. *Big Data & Society*, 3(2):2053951716665128, 2016.
- [3] A. Dubey, N. Naik, D. Parikh, R. Raskar, and C. A. Hidalgo. Deep learning the city: Quantifying

-
- urban perception at a global scale. *arXiv preprint arXiv:1608.01769*, 2016.
- [4] M. Foth. *Handbook of research on urban informatics: The practice and promise of the real-time city*. Information Science Reference Hershey, PA, 2009.
- [5] M. Foth. Lessons from urban guerrilla placemaking for smart city commons. In *Proceedings of the 8th International Conference on Communities and Technologies*, pages 32–35. ACM, 2017.
- [6] M. Foth, J. H.-j. Choi, and C. Satchell. Urban informatics. In *Proceedings of the ACM 2011 conference on Computer supported cooperative work*, pages 1–8. ACM, 2011.
- [7] R. Kitchin. The ethics of smart cities and urban science. *Philosophical Transactions of the Royal Society A: Mathematical, Physical and Engineering Sciences*, 374(2083):20160115, 2016.
- [8] R. Kitchin. Thinking critically about and researching algorithms. *Information, Communication & Society*, 20(1):14–29, 2017.
- [9] C. Norberg-Schulz. *Genius Loci: Towards a Phenomenology of Architecture*. Rizzoli, 1980.
- [10] G. Patterson and J. Hays. Sun attribute database: Discovering, annotating, and recognizing scene attributes. In *2012 IEEE Conference on Computer Vision and Pattern Recognition*, pages 2751–2758. IEEE, 2012.
- [11] D. Quercia, L. M. Aiello, R. Schifanella, and A. Davies. The digital life of walkable streets. In *Proceedings of the 24th international conference on World Wide Web*, pages 875–884. International World Wide Web Conferences Steering Committee, 2015.
- [12] B. Zhou, A. Khosla, A. Lapedriza, A. Torralba, and A. Oliva. Places: An image database for deep scene understanding. *arXiv preprint arXiv:1610.02055*, 2016.